# Histamine and Tyramine in Chihuahua Cheeses during Shelf Life: Association with the Presence of tdc and hdc Genes

**DOI:** 10.3390/molecules28073007

**Published:** 2023-03-28

**Authors:** Eduardo Campos-Góngora, María Teresa González-Martínez, Abad Arturo López-Hernández, Gerardo Ismael Arredondo-Mendoza, Ana Sofía Ortega-Villarreal, Blanca Edelia González-Martínez

**Affiliations:** Universidad Autónoma de Nuevo León, Centro de Investigación en Nutrición y Salud Pública, Facultad de Salud Pública y Nutrición, Monterrey 64460, Nuevo León, México

**Keywords:** Chihuahua cheese, histamine, tyramine, hdc and tdc genes, HDC and TDC proteins

## Abstract

Cheese is a product of animal origin with a high nutritional value, and it is one of the most consumed dairy foods in Mexico. In addition, Chihuahua cheese is the most consumed matured cheese in Mexico. In the production process of Chihuahua cheese, maturation is carried out by adding acid lactic microorganisms, mainly of the *Lactococcus* genus and, in some cases, also the *Streptococcus* and *Lactobacillus* genus. As part of the metabolism of fermenting microorganisms, biogenic amines can develop in matured foods, which result from the activity of amino decarboxylase enzymes. In cheeses, histamine and tyramine are the main amines that are formed, and the consumption of these represents a great risk to the health of consumers. In this work, the presence of biogenic amines (histamine and tyramine) was determined by HPLC at different times of the shelf life of Chihuahua cheeses. In addition, the presence of genes hdc and tdc that code for the enzymes responsible for the synthesis of these compounds (histidine and tyrosine decarboxylase, or HDC and TDC) was determined by molecular techniques. A significant correlation was observed between the presence of both histamine and tyramine at the end of shelf life with the presence of genes that code for the enzymes responsible for their synthesis.

## 1. Introduction

Cheese is a dairy product with a high nutritional value. Since it is one of the most consumed foods of animal origin, it is part of the basic food basket in several Latin American countries. In Mexico, a wide variety of cheeses from different regions of the country (Panela, Oaxaca, Cotija, Asadero, Chihuahua, Sierra, etc.) are consumed, with fresh cheeses being the most consumed. Within the group of matured cheeses, the most consumed is Chihuahua or Mennonite cheese, so-called because it was originally made by this resident community of the state of Chihuahua. Although this type of cheese originated in Chihuahua, it is currently prepared by different companies throughout the country, and it is also exported to the United States to meet the demands of the Hispanic community [1,2].

In the dairy industry, the starter cultures used to produce fermented products (cheese, cottage cheese, butter, kefir, yogurt, etc.) are microorganisms that are used to ferment glucose, transforming it into lactic acid. The microorganisms that are most used in these processes are lactic acid bacteria (LAB), whose main characteristic is that they cause a decrease in pH, which inhibits the growth of other microorganisms in fermented food, also promoting the coagulation of casein [3]. Starter cultures grow from the beginning of inoculation until reaching very high cell densities (10^8^–10^9^ CFU/mL) in the first hours of fermentation. Subsequently, there is a gradual decrease throughout the cheese maturation process. Although starter cultures can be made up of different types of microorganisms, mesophilic bacteria cultures of the genera *Lactococcus, Lactobacillus,* and *Leuconostoc*, or thermophilic species of *Lactobacillus,* such as *L. delbrueckii, L. helveticus*, or *Streptococcus thermophilus*, are generally used for the maturation of cheeses [4,5].

During cheese maturation, LABs are not only responsible for the production of lactic acid, but they also participate in proteolysis by hydrolyzing polypeptide chains and releasing amino acids that serve as precursors for the formation of volatile compounds largely responsible for the organoleptic properties of cheese [6]. The amino acids produced are catabolized by an initial pathway in which an aminotransferase enzyme acts, followed by two catabolic pathways where deamination or decarboxylation occurs. While deamination generates aromatic products due to the action of dehydrogenases (α-ketoacid and ammonia) or oxidases (aldehyde and ammonia), by-products of decarboxylation of biogenic amines can be generated (histamine, tyramine, putrescine, cadaverine, etc.). Some of these compounds can cause adverse physiological effects in susceptible consumers [7,8,9].

The activity of decarboxylase enzymes depends on the genetic potential, as well as other factors such as the pH of food (slightly acidic), availability of free amino acids, presence of fermentable carbohydrates (glucose), presence of pyridoxal phosphate (cofactor of the decarboxylation reaction), maturation temperatures, humidity, and a low concentration of salts, which favors the action of bacteria with amino decarboxylase enzymes [10,11,12]. Although temperatures above 15 °C favor the development of microorganisms and, therefore, of biogenic amines, the activity of the amino decarboxylase enzyme has been reported at temperatures of 4 °C [13,14,15,16].

The formation of histidine and tyramine (the biogenic amines that present a greater health risk) during the metabolism of LABs used in the elaboration of cheese is due to the presence of histidine decarboxylase and tyrosine decarboxylase, the enzymes responsible for their synthesis encoded by the hdc and tdc genes, respectively. Under normal conditions, the concentrations of biogenic amines in food do not present a health risk because the digestive system of the human organism has an efficient mechanism to eliminate these molecules through the enzymes monoamine oxidase (MAO) and diamine oxidase (DAO). However, under specific conditions, such as high concentrations of amines or in individuals that present deficiency or inhibition of these enzymes, there may be some risk of adverse reactions [17]. Symptoms of histamine poisoning occur in manifestations of the digestive system, such as nausea, vomiting, and diarrhea, in addition to hypotension, edema, tachycardia, headaches, and migraines [18,19].

Fermented cheeses have also been linked to severe hypertensive crises in patients who take medications that inhibit the enzyme monoamine oxidase (MAO). The amine related these events is tyramine, therefore, these clinical manifestations are called “cheese reactions” [20,21]. Although the symptoms appear from within a few minutes to three hours of ingestion of the contaminated product, it has been described that its duration depends, in large part on: (1) The physiological condition of the patient, reestablishing in hours or up to several days; and (2) The concentrations of histamine and other amines ingested with food [22]. On the other hand, it has been reported that an intake of 5–10 mg of histamine can be considered a risk factor for sensitive people, from 10 to 100 mg is considered a tolerable limit, from 100 to 1000 mg produces a medium intoxication, and amounts greater than 1000 mg produce severe intoxication [23]. Despite the toxic effects related to the biogenic amines, histidine and tyramine, in Mexico, there is no regulation of the allowable limits of these amines in cheeses. It is noteworthy that the production of biogenic amines is carried out mainly during the maturation process of Chihuahua cheese, which covers an approximate period of two months at temperatures between 4 and 27 °C. The production of these compounds continues during the storage of the cheese, usually at refrigeration, between 10 and 20 °C [24,25].

The aim of this study was to evaluate the concentration of histamine and tyramine using High-Performance Liquid Chromatography (HPLC) in Chihuahua cheeses at different times of their shelf life, and to determine the presence of the genes that code for the enzymes responsible for the synthesis of these compounds in cheese samples using molecular techniques (Polymerase Chain Reaction: PCR).

## 2. Results

### 2.1. Microbiological Quality of the Cheese Samples

Results corresponding to the microbiological quality of cheeses are presented in Table 1. As expected in matured products, all the cheeses have a high LAB content (ranging from 7.6 to 9 Log CFU/g). The total count of aerobic mesophilic bacteria was found between 3.1 and 3.8 Log CFU/g, and the total coliform bacteria was between 2.6 and 3.4 Log CFU/g. Statistical analysis showed a significant difference between the different brands.

At the end of the shelf life, the microorganisms recognized as histamine and tyramine producers were identified by API biochemical system. The identification of bacteria was performed up to the species level; the results are presented in Table 2. *L. pentosus* was found in three different brands, while *L. rhamnosus* was found in two brands.

### 2.2. Histamine and Tyramine Quantification

The biogenic amine quantification was carried out by the HPLC method with a fluorescence detector (FLD). The retention times for histamine and tyramine were 15 and 30 min, respectively. Calibration curves carried out with the standards showed linearity with different concentrations (25, 50, 100, 250, and 400 mg/L) of compounds. A correlation coefficient of 0.993 was determined for histamine and 0.987 for tyramine. Based on these values, a limit of detection (LOD) of 25 mg/L was determined for both amines under the experimental conditions of this study.

HPLC analysis was performed at three moments in the shelf life of cheeses (Table 3). In the first week of its acquisition (beginning of the shelf’s life), the presence of histamine was not detected in any of the cheeses, while tyramine was present in 37.5% of the cheeses (Cheeses B, C, and F). Tyramine presence reached concentrations ranging from 34–122 mg/kg. At half of shelf life, the presence of histamine was detected only in one of the cheeses (12.5% of cheeses), while tyramine was detected in 62.5% of the cheeses (five out of eight). Interestingly, it was observed that the concentration of tyramine in Cheeses B, C, and F (in which tyramine was detected from the first analysis) increased significantly to reach values of 160, 160, and 139 mg/kg, respectively. At this stage, in Cheese A, both histamine and tyramine were detected at concentrations between 45 mg/kg and 59 mg/kg, respectively.

At the end of the shelf life (approximately four months after its acquisition) histamine was detected in three of the eight cheeses (A, D, and G), which corresponds to 37.5% of them. In Cheese A, histamine concentration increased by more than 400% (from 45 to 192 mg/kg). On the other hand, tyramine was detected in 75% of cheeses (six out of eight) in concentrations from 115 mg/kg to 209 mg/kg.

It is noteworthy that while the concentration of tyramine increased in Samples B, C, D, and F over time, in Cheeses E and H, the presence of histamine and tyramine was not detected at the different storage times analyzed.

### 2.3. Detection of hdc and tdc Genes in Cheese Samples

For the determination of bacterial DNA in the cheese samples, the first-stage PCR amplification was performed using the specific oligonucleotides PO (GAGAGTTTGATCCTGGCTCAG) and 338-F (GCTGCCTCCCGTAGGAGT), which were reported by Ventura, et al., (2001) [26]. These oligonucleotides allow for the amplification of a 332 base-pair fragment corresponding to the gene that encodes the 16S ribosomal subunit of eubacteria. In all the cheeses analyzed, the expected PCR products were obtained (data not shown), indicating the presence of bacterial DNA in all cheese samples.

For the specific analysis of hdc gene presence, specific oligonucleotides (HDC1: TTGACCGTATCTCAGTGAGTCCAT and HDC2: ACGGTCATACGAAACAATACCATC), which were designed and reported by Fernandez et al., (2006) [27], were used. In addition, the obtained PCR products (bands of 174 bp) were found only in three of the samples (Cheeses A, D, and G), showing that 37.5% of cheeses analyzed contain bacteria able to produce HDC (Figure 1A).

On the other hand, to test the usefulness of degenerated primers designed to specifically amplify the tdc gene in LAB, DNA obtained from the cells of *L. brevis* (reference strain) was used as a template (Figure 1B, Line CTL+). PCR reactions using the degenerate oligonucleotides TER-F (GCWAAYYTDGARGGDYTHTGGTATGC) and TER-R (CCAWGAATARTGYTTHGTTTGTGG) were performed at different alignment temperatures (50–60 °C). As PCR products, DNA bands of the expected size (252 bp) were obtained. In six of eight cheeses (A, B, C, D, F, and G), the corresponding band to the tdc gene was observed, suggesting the presence of bacteria carrying the tdc gene, in 75% of the cheeses analyzed (Figure 1B).

## 3. Discussion

Chihuahua cheese is the matured cheese more consumed in Mexico and, in general, in several Latin American countries. Lactic acid bacteria are used in the Chihuahua cheese elaboration. However, LAB can produce biogenic amines during the process, causing adverse physiological effects in cheese consumers [9]. In Mexico, there is no regulation about the permitted levels of these compounds. In addition, there is no standardized process allowing the biogenic amines identification or species capable of producing them in this type of food.

Biogenic amine concentration can vary between cheeses since it depends on different factors during its elaboration: the quality of the raw material, the presence of strains used as starters, or the presence of contaminating microorganisms with amino decarboxylase activity. It has been reported that certain bacteria produce biogenic amines: *L. buchneri, L. fermentum, L. helveticus, L. rhamnosus L. brevis* [28], *L. fermentum, L. plantarum, L. helveticus* [29], *L. sakei,* and *L. pentosus* [30]. These species are commonly found in cheeses and are used as starter lactic acid bacteria cultures. On the other hand, both histamine and tyramine formations in cheeses have also been related to the presence of different bacterial species known as non-initiating lactic acid bacteria, mainly *Lactobacillus curvatus* and *L. lactis lactis*, respectively [28,30,31].

In this work, it has been demonstrated that the presence of biogenic amines in samples (histamine and tyramine) of Chihuahua cheese at different storage times. During the first week of shelf life, the histamine concentration was low or undetectable by the HPLC methodology. On the other hand, tyramine was found in concentrations ranging from 34–122 mg/kg in three of the eight samples. During the mid-shelf-life analysis, histamine and tyramine were detected in 12.5% and 62.5% of the samples, respectively. In the final tests (end shelf life), 37.5% of the cheeses were positive for the presence of histamine, and 75% for tyramine.

These results suggest that biogenic amine production takes place not only during the maturation process of the cheeses, but that such compound production continues during their shelf life. This phenomenon has already been reported by Diaz–Cinco et al., (1992) [32], who stated that cheese samples in their study were stored between 5–25 °C for 12 days, and the concentrations of histamine and tyramine increased both in higher temperatures and longer storage times. It is noteworthy that samples in our study were always kept at 1 °C during the storage period. This could indicate that even when cheeses are stored at low temperatures, the production of amines continues. The effect of low temperatures is well-known in delaying the growth rate of several bacterial microorganisms. However, such a condition is probably not adequate for the inactivation of enzymes responsible for amine production.

Histamine and tyramine concentrations in the different cheese samples showed marked variations despite belonging to the same type of cheese. These variations could be because cheeses were made by different manufacturers who probably used different starter cultures (or different strains), variations in the production process, and conservation of raw material.

In previous studies on matured cheeses, tyramine has been highlighted as the most frequent of the biogenic amines [33,34,35]. Similarly, in this study, tyramine was detected in six of the eight analyzed samples at the end of shelf life, while histamine was detected only in three of eight cheese samples. This biogenic amine is considered of particular interest due to the vasoactive effect it produces on susceptible consumers who have inhibition of MAO (monoamine oxidase enzyme). It has been described that the intake of 6 mg of tyramine showed mild effects, while the intake of 10–25 mg increased the risk of hypertension when ingested in combination with MAO inhibitors [36].

Despite the well-documented toxic effects of biogenic amines, current regulations in Mexico related to the maximum concentration of histamine only exist for fresh and processed fish, and there are no regulations for tyramine. According to the Norma Oficial Mexicana (NOM-242-SSA1-2009) [37], the limit of histamine is 100 mg/kg, which is similar to the value established by the European Union for fermented foods [38]. Within this margin, only a sample of Chihuahua cheeses analyzed in this study did not comply with this regulation. The Food and Drug Administration (FDA, Silver Spring, MD, USA) sets a maximum limit of 50 mg/kg for histamine in foods and considers of risk the concentrations between 50–200 mg/kg [39]. According to this, it could be stated that at the end of shelf life, 37% of Chihuahua cheeses may be a health risk (since three cheese samples presented higher concentration levels).

Using degenerated primers allowed for the detection of bacterial species capable of producing biogenic amines along the shelf life of cheeses. The presence of these LAB species at early shelf life could not be detected by routine microbiological methods, probably because the number of bacteria was not enough for its detection by these methods. However, such bacteria quantity could be detected by PCR due to the detection limit and sensibility this technique possesses. In our results, the use of degenerated primers agrees with the detection of tyramine by the HPLC technique (six of eight cheese samples), but it is interesting to note that the HPLC detection in six samples was performed only at the end of the shelf life.

## 4. Materials and Methods

### 4.1. Sample Origin

Eight Chihuahua kinds of cheese of different brands marketed in the Monterrey, N.L., Mexico metropolitan area were selected and analyzed. Cheese samples were kept at refrigeration temperature (2–8 °C) during transportation, and they were kept refrigerated at 1 °C throughout the study. A one-letter code was assigned to each sample. Relevant information was recorded (the elaboration and expiration date and the production batch). Three samples of each cheese were selected, and the analyses were carried out in triplicate.

### 4.2. Microbiological Analyses

Microbiological analyses were carried out during the first week of the acquisition of the cheeses. The analyses were performed according to the criteria of the Norma Oficial Mexicana (NOM-121-SSA1-1994) [40], which establishes sanitary specifications for fresh, matured, and processed cheeses, and the Norma Mexicana (NMX-F-209-1985) [41], which establishes specifications of the product named “Chihuahua-type cheese”.

The aerobic mesophilic bacteria count was performed by seeding in trypticase soybean agar, and samples were incubated at 37 °C for 24 h. For the coliform bacteria cultures, the specific culture medium Rida^®^ COUNT coliform (R-Biopharm AG Darmstadt, Germany) was used; the cultures were incubated for 24 h at 35 °C, and the presence and count of *Escherichia coli* were determined using Petrifilm *E. coli*/Coliform Count Plate (3M, Minnesota, USA) following the procedure recommended by the manufacturer. The count of LAB was made using cheese samples of 10 g and were homogenized with 90 mL of saline solution (0.85%) in sterile jars. Next, each bacterial culture was serially diluted (10^−1^–10^−10^) and pour-plated onto MRS agar and incubated at 37 °C for 48 h under anaerobic conditions using Gas-Pack-System (BD, Ontario, Canada), as recommended by [42]. The determination of LAB was performed in triplicate for each sample, and the identification was performed using the API^®^ (BioMérieux, Marcy-l’Étoile, France) biochemical system.

### 4.3. Histamine and Tyramine Analysis

The determination and quantification of the biogenic amines (histamine and tyramine) were carried out in three stages of the cheeses’ shelf life: start (during the first three days after the acquisition); half (in the middle of its shelf life, considering the product label statement); and end (at the end of the shelf life, regarding the expiration date declared on the label). For the amine extraction from cheese samples, the methodology described by Elsanhoty, Mahrous, and Ghanaimy (2009) [43] was followed. Briefly, samples (10 g of cheese) were mixed with 10 mL of 10% trichloroacetic acid solution and homogenized for 15 min (Ultra-turrax homogenizer^®^, Daigger, IL, USA); the products obtained were centrifuged 10 min at 3000 rpm at 4 °C (Eppendorf, model 5804 R, Hamburg, Germany). The supernatant was filtered (Whatman Paper No. 1), transferred to 15 mL polypropylene tubes (Corning, NY, USA), and kept at −20 °C until use.

The quantification of histamine and tyramine was performed following the official method 977.13 of AOAC (Association of Official Analytical Chemists). This method is sensitive for the identification and quantification of histamine in seafood and consists of the extraction of the compound, the formation of a derivative with *o*-phthaldialdehyde (OPA), and the detection of the fluorescence developed [44]. For the formation of the fluorescent derivatives, 300 μL of cheese sample diluted in 120 μL of 0.4 M borate buffer solution were added 5 min before to the HPLC analysis with 120 μL of o-phthaldialdehyde reagent (200 mg of OPA mixed in 9 mL of methanol, 1 mL of 0.4 M sodium borate, pH 10, and 160 μL of 2-mercaptoethanol) (Sigma–Aldrich, Saint Louis, MO, USA).

The fluorescent derivatives were separated by HPLC (high-performance liquid chromatography) for amine quantification (Thermo Scientific, Spectra System, Waltham, MA, USA). The HPLC System was equipped with a fluorescence detector fixed to 338 nm and 430 nm (absorption and emission wavelength, respectively). A Waters Ultrasphere ODS 5 μm (4.6 × 250 mm) C18 column was set, and a circumvention gradient formed by 12.5 mM of phosphate buffer, pH 6.5 (Eluent A), and acetonitrile (Eluent B) was used. The gradient started with an 85:15% ratio of Solvents A and B, respectively, and a flow of 0.9 mL/min during 3 min; subsequently, Solvent B was increased to 60% with a final flow rate of 1.3 mL/min (3–24 min) and then returned to 15% of Solvent B (24–50 min) and remained isocratic for five more minutes.

Histamine dihydrochloride and tyramine hydrochloride (Sigma–Aldrich, Saint Louis, MO, USA) were used as standards for the identification and quantification of these compounds in the cheese samples. Standard solutions were prepared at different concentrations (25, 50, 100, 250, and 400 mg/L), and a standard calibration curve was constructed. For the HPLC analysis, 10 μL of standards and samples were injected.

### 4.4. Molecular Biology Techniques Standardization

Standard techniques were used for the molecular analysis (DNA extraction and PCR analysis). To establish the molecular protocols that would allow the identification of bacteria-producing biogenic amines, the reference strains ATCC 33222 of *Lactobacillus* 30a and ATCC 367 of *Lactobacillus brevis*, were used as controls. For preservation and experimental conditions, both strains were grown in MRS broth for 48 h and were kept stirring at 200 rpm. The *Lactobacillus* 30a culture was performed at 37 °C under anaerobic conditions, while the *L. brevis* culture was performed under aerobic conditions at 30 °C.

#### 4.4.1. DNA Extraction

DNA extraction from reference strain cultures was carried out following the technique described by Hoffman and Winston, (1999) [45], in which the cell wall is broken both by mechanical fractionation with glass beads and by chemical lysis using a buffer solution (TSNT buffer: 2% Triton, 1% SDS, 100 mM NaCl, 10 mM Tris-HCl, pH 8.0) followed by a DNA purification step with a phenol–chloroform mixture (50:50) and subsequent precipitation of DNA with 3 mM of sodium acetate and ethanol (96%).

For the direct extraction of bacterial DNA present in cheeses, an adaptation of the technique described by Randazzo et al., (2002) [46] was used. DNA purification was performed using the SV Total RNA Isolation System kit (Promega, Madison, WI, USA). Briefly, cheese samples (5 g) were placed in sterile dilution jars (130 mL with screw cap) containing 40 mL of a sterile solution of sodium citrate (2%) and were incubated at 45 °C for 30 min. Subsequently, 200 μL of sterile glass beads (0.1 mm diameter) were added, and the mixture was stirred manually for 5 min and left to stand for 10 min at room temperature. Next, 1 mL of the supernatant was transferred to a sterile 1.5 mL polypropylene tube and centrifuged for 3 min at 12,000 rpm (Eppendorf mini spin centrifuge; Eppendorf; Hamburg, Germany); the supernatant was eliminated. This process was repeated until the pellet product of all volumes (10 mL) of the homogenized cheese was obtained. From this product, the DNA was obtained using the SV Total RNA Isolation System kit, following the instructions from the manufacturer; the obtained DNA was diluted with sterile water (50 μL) and stored at −20 °C until use.

#### 4.4.2. Amplification of Genes Encoding hdc and tdc

The presence of the genes encoding decarboxylase enzymes (histidine and tyrosine) on DNA obtained from cheese was determined by PCR amplification. PCR results obtained from DNA amplification from different cheese samples were considered positive when PCR reactions were positive in the three representative cheese samples. For the identification of hdc and tdc genes in cheese samples, the conditions described by Fernandez et al., (2006) [27] were used. For the hdc gene amplification, the specific oligonucleotides HDC1 and HDC described by Ventura et al., (2001) [26] were used. For the tdc gene identification, a pair of degenerate oligonucleotides (TER-F and TER-R) was designed on the sequences from the tdc gene corresponding to different bacterial species.

#### 4.4.3. Design of Degenerated Oligonucleotides for the tdc Gene

Nucleotide sequences corresponding to genes that code for the enzyme tyrosine decarboxylase from different species of bacteria (*Enterococcus hirae*, *E. durans*, *E. faecium*, *Lactobacillus curvatus*, *L. brevis and Streptococcus* sp.) were obtained from databases. Alignments of these sequences (which allowed identifying the conserved regions) were made (both the amino acids and nucleotides levels) using the BLAST (http://blast.ncbi.nlm.nih.gov/Blast.cgi, accessed on 10 July 2020) and CLUSTAL-W (http://www.genome.jp/tools/clustalw/, accessed on 10 July 2020) programs. The regions that presented higher conservation were identified and used to design the degenerate oligonucleotides TER-F (forward primer) and TER-R (reverse primer).

The standardization of the experimental conditions of PCR amplification and determination of the specificity of both the specific oligonucleotides for hdc and the degenerate ones for the tdc genes was carried out using DNA obtained from the reference strains ATCC 33222 of *Lactobacillus* 30a and ATCC 367 of *L. brevis*, respectively. PCR reactions were performed in a Thermocycler (Axygen, Therm-1000, Maxygene) using a temperature gradient (50–60 °C) as an alignment/hybridization temperature. Final PCR conditions were 95 °C for 5 min (a cycle), followed by 95 °C for 45 s, 50–60 °C for 45 s, and 72 °C for 2 min (35 cycles), with a final extension cycle (72 °C for 5 min). The products obtained from PCR reactions were analyzed by electrophoresis in 1.5% agarose gels, stained with ethidium bromide [47], and were visualized under UV light with a Gel-Doc Photo Documentation System (Vision Works, UVP, Upland, CA, USA).

### 4.5. Statistical Analysis

Statistical analyses were performed using the statistical software package IBM SPSS Statistics V21.0. Differences between treatments were analyzed using the analysis of variance (ANOVA) test. The statistical significance was determined at *p* < 0.05. Data corresponding to each variable are expressed as means ± SD, unless otherwise indicated. All determinations were performed by triplicate.

## 5. Conclusions

In this study, the presence of biogenic amines, histamine and tyramine, was detected in 37% and 75%, respectively, at the end of the shelf life of the Chihuahua cheese tested. It was observed that at this point, the number of samples positive for the presence of histamine and tyramine increased, suggesting that even at cooling temperature, the production of these compounds continues. The presence of genes encoding for hdc and tdc was detected by molecular techniques using degenerated primers. The detected tyramine levels (<600 mg/kg) do not present a risk to healthy consumers (without MAO inhibitors prescription); in the other hand, the histamine content at the end of the shelf of cheeses (>50 mg/kg) represent a health risk, according to EFSA and FDA regulations. The presence of biogenic amines in foods, such as Chihuahua cheese, could be controlled by the strict use of good manufacturing hygiene practices and raw material selection.

In Mexico, there is no regulation about the maximum concentration of biogenic amines (histamine or tyramine) in cheeses. Although there are standardized protocols for amine detection, these processes are time-consuming and labor-intensive. In this work, we presented evidence of the usefulness of degenerated primers to be employed as tools for amplification by PCR techniques to achieve a direct detection of bacterial species that produce biogenic amines.

Moreover, we suggest the appliance of molecular techniques as a detection tool for bacteria that produce biogenic amines in food as it can prevent the early development of these bacteria and, therefore, biogenic amines in food. In addition, maximum regulatory limits on biogenic amines must be declared to protect consumers and avoid possible public health problems.

## Figures and Tables

**Figure 1 molecules-28-03007-f001:**
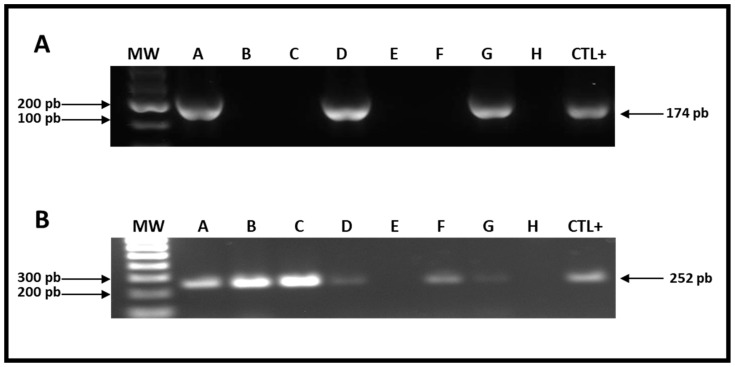
PCR amplification of histidine decarboxylase (**A**) and tyrosine decarboxylase (**B**) genes in LAB present in Chihuahua-type cheeses. (**A**) Amplification of histidine decarboxylase gene. Oligonucleotides HDC1 and HDC2 were used to amplify a 174-bp fragment of hdc gene in the bacterial DNA present in the different cheese samples (line A-H); genomic DNA corresponding to *Lactobacillus* sp. 30a strain was used as positive PCR-control (lane CTL+). (**B**) Amplification of tyrosine decarboxylase gene. Degenerated oligonucleotides TER-F and TER-R were used to amplify a 252-bp fragment of tdc gene from bacterial DNA present in the different cheese samples (line A-H) genomic DNA corresponding to *L. brevis* strain was used as positive PCR-control (lane CTL+). DNA molecular weight marker was included in the electrophoresis analyses (line MW).

**Table 1 molecules-28-03007-t001:** Microbiological quality of Chihuahua cheeses.

Cheeses	Aerobic Mesophilic Log CFU/g	Total Coliforms Log CFU/g	LAB Log UFC/g
A	3.7 ± 0.03 ^a^	3.3 ± 0.01 ^c^	9.0 ± 0.02 ^a^
B	3.1 ± 0.02 ^b^	3.1 ± 0.02 ^a^	8.3 ± 0.04 ^c^
C	3.1 ± 0.01 ^b^	3.3 ± 0.02 ^c^	8.3 ± 0.04 ^c^
D	3.6 ± 0.01 ^a^	3.3 ± 0.01 ^c^	8.6 ± 0.04 ^c^
E	3.7 ± 0.02 ^a^	2.6 ± 0.01 ^b^	7.6 ± 0.03 ^e^
F	3.4 ± 0.03 ^c^	3.3 ± 0.01 ^cd^	8.8 ± 0.04 ^b^
G	3.8 ± 0.02 ^e^	3.4 ± 0.02 ^d^	7.8 ± 0.03 ^d^
H	3.8 ± 0.02 ^e^	3.3 ± 0.03 ^cd^	7.8 ± 0.03 ^d^

Capital letters represent each one of included cheeses. Values correspond to average ± standard deviation of Log CFU/g in three independent experiments. Different superscripts ^a–e^ indicate significant differences between cheese brands (*p* < 0.05). Log CFU/g: Logarithm of Colony Forming Unit per gram. LAB: Lactic Acid Bacteria.

**Table 2 molecules-28-03007-t002:** Biochemical identification of strains recognized as histamine and tyramine producers in the Chihuahua cheeses.

Cheeses	HDC	TDC
A	*L. pentosus*	*L. pentosus*, *L. plantarum*
B	ND	ND
C	ND	*L. pentosus*
D	ND	*L. lactis sub lactis*
E	ND	ND
F	ND	*L. rhamnosus*
G	*L. pentosus*	*L. rhamnosus*
H	ND	ND

Identification of bacteria based on API biochemical tests at the end of the shelf life. Capital letters represent each different included cheese. HDC: histidine decarboxylase; TDC: tyrosine decarboxylase; ND: bacteria were not detected.

**Table 3 molecules-28-03007-t003:** Detection of biogenic amines in Chihuahua cheeses.

Cheese	Histamine (mg/kg)			Tyramine (mg/kg)		
	Shelf Life			Shelf Life		
	Start	Half	End	Start	Half	End
A	ND	45 ± 4	192 ± 20	ND	59 ± 15	115 ± 10
B	ND	ND	ND	122 ± 12	160 ± 18	209 ± 2
C	ND	ND	ND	34 ± 2	160 ± 17	166 ± 6
D	ND	ND	64 ± 11	ND	147 ± 4	194 ± 24
E	ND	ND	ND	ND	ND	ND
F	ND	ND	ND	42 ± 8	139 ± 9	150 ± 1
G	ND	ND	46 ± 6	ND	ND	205 ± 4
H	ND	ND	ND	ND	ND	ND

Biogenic amine detection was performed by HPLC at three moments in the shelf life of cheeses. Capital letters represent each one of included cheese. Values correspond to average ± standard deviation of amines concentration (mg/kg) determined in three independent experiments. mg/kg: correspond to amines quantity by kilogram of cheese.

## Data Availability

Not applicable.

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
