# Peer review of "Histamine and Tyramine in Chihuahua Cheeses during Shelf Life: Association with the Presence of tdc and hdc Genes"

_molecules, 2023, doi:10.3390/molecules28073007_

Round 1

Reviewer 1 Report

line 97: The aim of the work should start with the a new paragraph.

line 98: It should be high-perform liquid chromatography (HPLC).

line 109: What was the statistical analysis? What tests were used? There are no information in ,,Materials and Methods’’chapter. How to interpret superscript a-e?

line 124: It should be TDC: tyrosine decarboxylase.

lines 127-128: Incorrect formulation. It should be for example ,,The research was carried out by HPLC method with fluorescence detector (FLD). The retentions times for histamine and tyramine were 15 and 30 min, respectively’’.

line 129 and 319-320: The concentrations of the standards should be listed in order from lowest to highest (25, 50, 100, 250 and 400 mg/l).

line 155: It should be mg/kg.

line 213: Unnecessary parentheses.

line 246: No reference for Food and Drug Administration.

line 257 and 388: It should be tyramine instead tyrosine.

line 299: The method 954.04 of AOAC is titled ,,Histamine in seafood. Biological method.’’ I think this is a mistake. The authors used chemical methods to perform analysis but not biological.

lines 383-384: It should be: ,,In this study, the presence of biogenic amines: histamine and tyramine, were detected in 37% and 75%, respectively, at the end of the shelf life of the Chihuahua cheese tested’’.

lines 388-390: On what basis this conclusion was formulated? This is not found the confirmed in Table 3. The levels of tyramine was higher than histamine in this study.

Author Response

The suggestions were considered in the new version of paper Manuscript ID: molecules-2274910...

Thanks for your comments. The suggestions were attended, as indicated in the manuscript.

Reviewer 2 Report

This manuscript presents an original study about the assessment of histamine and tyramine concentration using high performance liquid chromatography (HPLC) in Chihuahua cheeses at different times of shelf life and the determination of the presence of genes encoding the enzymes responsible for the synthesis of these compounds in cheese samples using molecular techniques (polymerase chain reaction PCR).

The experimental data showed that the presence of biogenic amines: histamine and tyramine, were detected in 37% and 75%, respectively, of the Chihuahua cheese tested. It was observed that at the end of the shelf life, the number of positive samples for the presence of histamine and tyramine increased. The presence of genes encoding hdc and tdc was detected by molecular techniques using degenerate primers.

The scientific quality of the manuscript it rises to the scientific level of the Molecules Journal. The technical quality of the manuscript is good in terms of how it was written and how the experimental results are presented. The style of expression reflects the scientific training of the authors. The paper is edited in accordance with the article drafting requirements.

The Abstract is concise and contains sufficient information to highlight the content of the article and the Introduction section provides a clear statement of the problem studied in the present manuscript.

The Materials and Methods section is well presented and appropriate to the purpose of the research.

Results follow the guidelines described in the Author's Guide.

References are relevant and current and follow the journal’s format.

The conclusions of the article partially reflect the results of the given study.

Author Response

The suggestions were considered in the new version of paper Manuscript ID: molecules-2274910...

Thanks for your comments. The conclusions section was re-edited.
